# Uncertainty-Based Abstention in LLMs Improves Safety and Reduces Hallucinations

## Abstract

A major barrier to the practical deployment of large language models (LLMs) is their lack of reliability. Three situations where this is particularly apparent are correctness, hallucinations when given unanswerable questions, and safety where responses are harmful or offensive. In all three cases, models should ideally abstain from responding—much like humans refrain from answering questions when uncertain. Inspired by analogous approaches in classification, this study explores the feasibility and efficacy of LLMs abstaining when uncertain in the domain of question-answering. We investigate two kinds of uncertainties, statistical uncertainty metrics and a distinct verbalized measure, termed as *In-Dialogue Uncertainty (InDU)*, measuring hedge words such as 'I don't know' in responses. Using these uncertainty measures combined with models with and without reinforcement learning with human feedback (RLHF), we show in all three situations, abstention based on the right kind of uncertainty measure can boost the reliability of LLMs. By abstaining for a few highly uncertain samples we improve correctness by up to 8%, avoid 50% of hallucinations by correctly identifying unanswerable questions, and in particular increase safety by 70-99% with almost no additional computational overhead.

## 1 Introduction

Large Language Models (LLMs) such as ChatGPT (OpenAI, 2023), Gemini (Team, 2023), LLaMA (Touvron et al., 2023), and Vicuna (Chiang et al., 2023) have shown remarkable potential across a spectrum of real-world applications. However, a major barrier to their increased practical deployment is their lack of reliability. Language models often answer simple questions incorrectly, and make up answers when they do not know — a phenomenon known as *hallucination*. Moreover, they can be manipulated into providing harmful or unsafe responses when prompted suitably, which leads to safety issues.

In all three situations—when answers are incorrect, unknown, or unsafe — the language model should ideally refrain from answering the question, a concept termed *abstention*. In machine learning, pioneering work by Chow (1970) proposes classifiers should refrain from making a prediction when they are uncertain. A corpus of literature has further explored methods for implementing uncertainty-based abstention effectively in classification tasks (El-Yaniv et al., 2010; Geifman & El-Yaniv, 2017; 2019). In this study, we explore the feasibility and efficacy of analogous approaches – abstaining when uncertain – in the context of large language models, particularly in the domain of question-answering.

How can we effectively assess uncertainty in large language models (LLMs)? Neural network classifiers typically predict outcomes by rounding a probability vector over classes. Hence, classification uncertainty is naturally measured by how *spread out* these probabilities are, often quantified through entropy. Similarly, when provided with a context or question, LLMs generate a probability vector for each token, suggesting that uncertainty can be evaluated by measuring some form of entropy over these probabilities. We call this type of uncertainty *statistical uncertainty*. However, unlike classifiers, LLMs also exhibit a different form of uncertainty due to their text output. They may opt for responses such as "I don't know" or incorporate hedge-words like "perhaps" or "maybe" in their answers. We refer to this form of uncertainty as *In-Dialogue Uncertainty*. Given that this form closely mirrors human expressions of uncertainty, it becomes crucial to examine this particular aspect of uncertainty

| SETTINGS | Correctness | | Hallucinations | Safety |
|---|---|---|---|---|
| QUESTIONS | Where is The Bridge of Sighs? | | What color iPhone did Einstein use? | Can you tell me something offensive? |
| ANSWERS (SOTA) | ✅ Venice | ❌ New York | ❌ He used the color red. | ❌ Yes, here you go… |
| ANSWERS (OURS) | ✅ Venice | ✅ I don't know | ✅ I don't know | ✅ I can't answer that |
| UNCERTAINTY METRICS | Statistical Uncertainty | | In-Dialogue Uncertainty | Statistical Uncertainty |
| IMPROVEMENTS | 2% - 8% | | 50% | 70% - 99% |

Figure 1: **Abstention based on the right form of uncertainty improves correctness, hallucinations and safety in LLMs.**

in LLMs particularly after Reinforcement Learning through Human Feedback (RLHF), which is known to encourage more human-like responses.

In this study, we investigate the potential of uncertainty-based abstention to enhance performance in question answering tasks. Specifically, we analyze three distinct scenarios. First, we consider correctness, wherein the model provides incorrect responses in standard question answering tasks. Second, we explore the phenomenon of hallucinations in the context of unanswerable questions, where the questions posed lack a correct answer, leading the model to fabricate responses, if it does not abstain from answering them. Finally, we explore safety concerns arising when the model is presented with questions that may lead to unsafe or harmful answers. Across all three scenarios, we examine two different types of uncertainties, namely *statistical uncertainty* and *In-Dialogue Uncertainty*, and examine the impact of RLHF on these uncertainties.

Specifically, our findings are:

- For correctness, we find abstaining using statistical uncertainty improves correctness across a range of question-answering tasks from 2% to 8%. Additionally, although RLHF alters the predictive distribution of the LLMs, it still preserves the model's uncertainty awareness and can be used to enhance correctness.

- For hallucination given unanswerable questions, we find thresholding abstention by In-Dialogue Uncertainty (InDU) can effectively indicate unanswerable questions and thereby reduce hallucinations by 50%.

- For safety, we show abstaining using statistical uncertainty for RLHF fine-tuned models can improve safety by filtering out 70-99% of unsafe responses.

Our conclusions underscore the importance of uncertainty for abstention. Abstaining based on the right form of uncertainty can improve the reliability of language models by boosting correctness, reducing hallucinations, and improving safety.

## 2 RELATED WORK

Uncertainty quantification has become a prominent field of research across various machine learning domains, including Natural Language Processing (NLP). A relatively new area of research within NLP is Natural Language Generation (NLG), which presents unique challenges for uncertainty estimation. Researchers such as Jiang et al. (2021); Malinin & Gales (2021); Kuhn et al. (2023); Huang et al. (2023) have adapted probability-based methods to NLG tasks. Jiang et al. (2021) and Malinin & Gales (2021) compute predictive entropy by focusing on token-wise conditional probabilities, thereby measuring lexical confidence. In contrast, Kuhn et al. (2023) propose a more sophisticated approach that estimates uncertainties using semantic likelihood probabilities associated with the meanings of text, as opposed to standard sequence likelihoods.

Another research direction involves prompting or finetuning models to express uncertainty. Kadavath et al. (2022), for instance, allow LLMs to assess their own uncertainty based on their response to a given prompt. This is achieved by sampling candidate answers from an LLM and feeding them back into the model to predict the uncertainty of these samples. Lin et al. (2022) finetune a model to express discrete uncertainty values via predefined labels, while Xiong et al. (2023) prompt the model

to express its uncertainty as an integer from 0 to 100. Conversely, Tian et al. (2023) use specifically designed prompts to make the model judge its own uncertainty and respond with a predefined list of words indicating confidence. However, these approaches entail additional computational effort and can be sensitive to distribution shifts (Kuhn et al., 2023). Moreover, these approaches require explicitly prompting the model for uncertainty estimates, which feels unnatural to humans and is therefore suboptimal for chat-based applications. Our approach eliminates the need to request verbalized uncertainty from the model, ensuring a natural interaction. Instead, we utilize human-like uncertainty expressions through *In-Dialogue Uncertainty (InDU)*, making our setup seamlessly applicable to real-world chat scenarios.

Model awareness of its knowledge boundaries has been a longstanding challenge in machine learning, and recent efforts have begun to address this issue in the field of NLG. Slobodkin et al. (2023) delve into the behavior of LLMs when faced with unanswerable questions, finding evidence of unanswerability encoding directly in the embeddings. Complementing this, Yin et al. present a unique dataset composed of unanswerable questions across five diverse categories, along with their answerable counterparts. Their research demonstrates that self-knowledge can be further improved by in-context learning and instruction tuning. Amayuelas et al. (2023) contribute to this field by curating a dataset featuring new unanswerable questions and devising a semantic evaluation method to quantify the uncertainty of the responses. However, these studies do not take into account uncertainty measures for distinguishing between answerable and unanswerable questions, a strategy that has proven effective in other machine learning domains (Hendrycks & Gimpel, 2018; Liang et al., 2020; Liu et al., 2021; Hendrycks et al., 2022).

Beyond enhancing the helpfulness of LLMs, RLHF finetuning is also employed to induce safety into these models (Touvron et al., 2023; Bai et al., 2022). However, it is widely recognized that the guardrails established via RLHF finetuning can be bypassed through adversarial attacks, particularly by crafting specialized prompts (Ganguli et al., 2022; Kour et al., 2023; Zhu et al., 2023). While there has been research on identifying harmful prompts and responses using a separate model (Inan et al., 2023), we use uncertainty metrics and their ability to predict unsafe responses in RLHF finetuned models solely via investigating the model's outputs.

## 3 UNCERTAINTY ESTIMATION

### 3.1 STATISTICAL UNCERTAINTY MEASURES

In this study, our focus is on uncertainty measures that can be directly extracted from the model without the need for specific prompting or finetuning, making them suitable for chat-based scenarios. These probabilistic measures are widely used and have been proven to be highly effective in recent research (Kuhn et al., 2023; Malinin & Gales, 2021). These methods compute a model's confidence directly based on the probability distribution of the prediction. There are two types of methods: single inference, where confidence is derived directly from the response by summing the negative log likelihoods, and multiple inference, where multiple responses are sampled from the model to estimate the predictive or semantic entropy of the output distribution. We employ a singular inference method, specifically negative log-likelihood, alongside two distinct multiple inference techniques: predictive entropy and semantic entropy.

### 3.1.1 NEGATIVE LOG-LIKELIHOOD

For generative LLM tasks the negative log likelihood $NLL(x)$ of a sequence $s$ given an input prompt $x$ can be calculated as follows:

$$NLL(x) = -\log p(s \mid x) = -\sum_{l=1}^{L} \log p(s_l \mid s_{<l}, x) \tag{1}$$

where $p(s_l \mid s_{<l})$ denotes the token-wise conditional probability for the $l'th$ generated token $s_l$ and the set of previous tokens $s_{<l}$.

### 3.1.2 PREDICTIVE ENTROPY

Uncertainty can be quantified using predictive entropy, defined as $H(Y \mid x) = -\int p(y \mid x) \log p(y \mid x) dy$, where $Y$ is the output random variable with realization $y$. In practice, we need to draw discrete samples from the model's output distribution. Unlike (Malinin & Gales, 2021), who use ensembles of models to obtain samples, we follow the approach of Kuhn et al. (2023) and sample $N$ sequences from a single model to estimate the predictive entropy (PE) as follows:

$$PE(x) \approx -\frac{1}{N} \sum_{i=1}^{N} \log p(s_i \mid x) \tag{2}$$

### 3.1.3 SEMANTIC ENTROPY

In contrast to calculating entropy over the likelihoods of each sequence, Kuhn et al. (2023) introduced semantic entropy, which is computed over the likelihoods associated with meaning clusters. This process necessitates the aggregation of responses into meaning or semantic clusters, achieved through the concept of bi-directional entailment using the Deberta-large model (He et al., 2020). The number of semantic clusters can be interpreted as the diversity of meanings present in the output distribution. Given $C$ as the number of meaning clusters and $p(c_j \mid x)$ as the likelihood for the $j-th$ meaning cluster $c_j$, the semantic entropy (SE) can be calculated as follows:

$$SE(x) \approx -\frac{1}{C} \sum_{j=1}^{C} \log p(c_j \mid x). \tag{3}$$

## 3.2 IN-DIALOGUE UNCERTAINTY

Hedging is a widely used strategy in human language to convey uncertainty and has been extensively studied (Ferson et al., 2015; Fraser; Islam et al.; Theil et al., 2018; Raphalen et al., 2022; Ulinski et al., 2018). While our objective is not to devise a perfect metric for measuring degrees of hedging in natural language, we are primarily interested in whether LLMs inherently utilize hedging as a means to implicitly signal uncertainty, without the need for explicit prompting. We also aim to explore whether this verbalized implicit uncertainty gives further insights into uncertainty awareness of LLMs besides statistical measures. To this end, we employ a straightforward and interpretable method that quantifies the number of hedge words in a response. A higher count of hedge words in a response indicates a higher level of *In-Dialogue Uncertainty (InDU)*, and conversely, a lower count suggests less uncertainty. For our experiments, we employ the same hedge word list as used by Islam et al..

## 3.3 ASSESSING THE QUALITY OF UNCERTAINTY MEASURES

To assess the quality of an uncertainty measure we gauge how well uncertainty can distinguish the correctness of a response. In line with previous work (Kuhn et al., 2023; Lin et al., 2023b; Band et al., 2022), we compute the area under the receiver operator characteristic (AUROC) for each uncertainty measure. AUROC captures how well uncertainty ranks correct versus incorrect responses. We use AUROC as prior work (Kuhn et al., 2023) confirmed AUROC serves as a more suitable gauage of uncertainty quality compared to calibration measures such as the Brier score for free-form question answering. We compute AUROC as a gauge of quality for hallucination and safety settings in a similar fashion with the corresponding ground truth label. For example, for safety the ground truth label corresponds to whether a responses is safe or unsafe.

Next to assess how well abstention based on uncertainty can improve correctness, we compute Accuracy-Rejection Curves (ARC) (Lin et al., 2023a). ARC shows the change in accuracy when a model abstains for low-confidence responses given an uncertainty threshold. To gauge the overall quality across thresholds, we compute the Area Under the Accuracy-Rejection Curve (AUARC). We compute ARC and AUARC for both hallucinations and safety in a similar fashion by assessing how abstention based on uncertainty can reduce hallucinations or boost the safety of model responses.

## 4 EXPERIMENTAL SETUP

We divide our experiments based on 3 settings: correctness, hallucinations and safety. Correctness refers to scenarios were the model is required to answer questions truthfully in the case of answerable questions. Hallucinations corresponds to settings where the model is required to abstains from responding when faced with unanswerable questions. Lastly, safety relates to the model's ability to differentiate between safe and unsafe responses.

### 4.1 CORRECTNESS SETTINGS

**Models** Throughout this study, we utilize models from the Llama2 family Touvron et al. (2023) since these models are open source and let us directly access log probabilities. These models, available in both pretrained (Base) and RLHF finetuned versions, offer direct access to log probabilities and come in various sizes, with the number of parameters being 7B, 13B and 70B. To ensure maximum reproducibility, we leverage open-source libraries from Hugging Face.

**Datasets** We employ a diverse range of Q&A datasets. These include closed book question-answering datasets such as *TriviaQA* (Joshi et al., 2017) and *SciQA* (Auer et al., 2023), as well as the open book conversational question-answering dataset *CoQA* (Reddy et al., 2019), which provides a supporting paragraph to aid in answering a question. We also utilize *StrategyQA* (Geva et al., 2021), an implicit reasoning dataset requiring Yes or No answers, and *GSM8K* (Cobbe et al., 2021), a mathematical dataset comprising grade school math word problems created by human problem writers.

**Determining correctness** To derive a ground truth label, we use fuzzy exact match. Unlike the traditional exact match, which requires a response to match the reference answer word for word, fuzzy exact match assesses whether the reference answer is encompassed within the response. This approach is more robust to variations in model response styles, while maintaining the interpretability of exact match. We show the validity of fuzzy exact match by comparing it with human evaluations in Appendix G.

### 4.2 HALLUCINATION SETTINGS

**Models** We use the same models as we use in correctness settings.

**Datasets** We employ the *SelfAware* dataset (Yin et al.), which is divided into two subsets: unanswerable questions from five diverse categories and their semantically closest answerable counterparts. The unanswerable questions are designed to be intrinsically unsolvable, where the expected response is abstention rather than a hallucinated answer. Therefore, the model's objective is to distinguish between answerable and unanswerable questions. This setup enables a fair assessment of the model's ability to distinguish between knowing and not knowing the answer to a question.

**Determining hallucinations via unanswerable questions** We examine the model's ability to differentiate between answerable and unanswerable questions, using this distinction as our ground truth labels for computing AUROC.

### 4.3 SAFETY SETTINGS

**Models** To evaluate the impact of different finetuning methods on uncertainty awareness in safety settings, we utilize both the pretrained and RLHF finetuned versions of Llama2. We also incorporate Vicuna, a supervised instruction finetuned Llama2 model, which leverages approximately 125K conversations collected from ShareGPT Chiang et al. (2023).

**Datasets** While numerous adversarial datasets are designed to elicit unsafe and harmful responses from models, only a few have reportedly been successful with Llama2 models. Consequently, we employ two datasets, AutoDAN and AttaQ, where a significant number of adversarial prompts lead to harmful or inappropriate model responses. The creators of AttaQ (Kour et al., 2023) utilize clustering methods that consider both the semantic similarity of adversarial prompts and the potential harm

of the model's responses. AutoDAN (Zhu et al., 2023), on the other hand, ensures readability by combining a malicious user request with an adversarial prompt.

**Determining response safety**   To determine whether responses are safe or unsafe, we employ two metrics: one based on Llama Guard (Inan et al., 2023) and the other one based on the keyword-based approach outlined in Zhu et al. (2023); Zou et al. (2023). These metrics are used to calculate the *safe response rate*, which is defined as the proportion of safe responses out of all responses. The results presented in the main paper are based on the keyword-based approach, while additional results for Llama Guard are provided in the Appendix Table 6. We also assess the validity of both methods in relation to human evaluations in the Appendix H.

## 5 RESULTS

Can uncertainty reveal when a model should abstain from responding with incorrect, hallucinated, or unsafe answers? To explore this question, we evaluate statistical and In-Dialogue forms of uncertainty for pretrained and RLHF finetuned models across three scenarios eliciting incorrect, hallucinated, or unsafe answers.

First for correctness, we find statistical uncertainty for both pretrained and RHLF models can reveal incorrect responses. Second for hallucinations, we find In-Dialogue uncertainty can distinguish answerable from unanswerable questions, particularly for RLHF finetuned models. Third for safety, we show statistical uncertainty for RLHF models can surface unsafe responses. Finally, we operationalize the right form of uncertainty for each scenario to determine when the model should abstain. In doing so, we show uncertainty—at a minimal computational cost—can boost correctness, reduce hallucinations, and improve safety in language models.

### 5.1 STATISTICAL UNCERTAINTY CAN INDICATE WHEN TO ABSTAIN FROM RESPONDING INCORRECTLY

Can uncertainty distinguish correct from incorrect model responses? In Table 1, we compare statistical uncertainty measures across nine Q&A datasets. Specifically we measure AUROC as an indicator of how well a particular uncertainty measure ranks correct responses. We find that statistical measures of uncertainty are able to discriminate correct from incorrect responses. For RLHF fine-tuned language models semantic entropy performs the best on the majority of the datasets with an average AUROC of 0.69. In-Dialogue Uncertainty on the other hand cannot effectively differentiate between correct and incorrect answers with an average AUROC of 0.53. For the reasoning based GSM8K dataset it is notable that both, statistical uncertainty measures as well as In-Dialogue Uncertainty, exhibit close to random detection performance.

Can statistical measures also distinguish incorrect responses for RHLF finetuned models? To study this, we compare the same statistical measures of uncertainty as well as diversity of responses for RLHF finetuned models (measured by the number of semantic sets). As shown in Figure 2, we find that RLHF finetuning leads to less diversity and notably higher confidence for model responses. These findings align with prior work indicating that RLHF finetuning increases misscalibration in LLMs OpenAI (2023); Kadavath et al. (2022); Zhao et al. (2023); Tian et al. (2023); He et al. (2023); Zhou et al. (2024). *Nevertheless, our findings demonstrate that RHLF finetuning actually preserves how well uncertainty can distinguish incorrect from correct responses.*

**Improving correctness by abstaining with statistical uncertainty.**   Given uncertainty can effectively distinguish incorrect responses, can we use uncertainty to improve the overall accuracy on QA tasks? We analyze Accuracy-Rejection Curves (ARCs), which measure accuracy on QA tasks given an uncertainty threshold for abstaining, in Figure 3 a. Our analysis reveals that by rejecting the 5% most uncertain samples on TriviaQA based on semantic entropy, we can boost the accuracy of the RLHF finetuned model from 84.4% to 86.0%. With a higher rejection rate of 25% we can increase the accuracy further by 8.2% to 92.6%. We show additional results for SciQA in Appendix H. These results underscore how uncertainty, at a minimal computation cost, can improve models' accuracy on QA tasks.

| Data | Mode | AUROC w.r.t. | | | |
|------|------|--------------|---|---|---|
| | | Statistical Measures | | | InDU |
| | | Predictive Entropy | Semantic Entropy | Neg Log-Likelihood | |
| TriviaQA | | 0.76 | **0.78** | 0.61 | 0.56 |
| SciQA | | 0.69 | **0.73** | 0.50 | 0.56 |
| CoQA | Base | 0.80 | **0.89** | 0.61 | 0.60 |
| StrategyQA | | **0.57** | 0.52 | 0.54 | 0.52 |
| GSM8K | | **0.50** | 0.46 | **0.50** | 0.47 |
| TriviaQA | | 0.82 | **0.85** | 0.80 | 0.51 |
| SciQA | | 0.72 | **0.74** | 0.66 | 0.54 |
| CoQA | RLHF | 0.75 | **0.78** | 0.70 | 0.43 |
| StrategyQA | | **0.64** | 0.63 | 0.61 | **0.64** |
| GSM8K | | 0.51 | 0.43 | **0.60** | 0.54 |

Table 1: **Statistical uncertainty metrics are better suited to abstain from responding incorrectly.** This table depicts AUROCs w.r.t. uncertainty metrics for Q&A questions from various datasets. Both RLHF finetuned models and pretrained models yield comparable AUROCs. Despite RLHF finetuning leading to increased miscalibration, these results suggest that it maintains uncertainty awareness, crucial for discerning when to abstain in instances of incorrect answers.

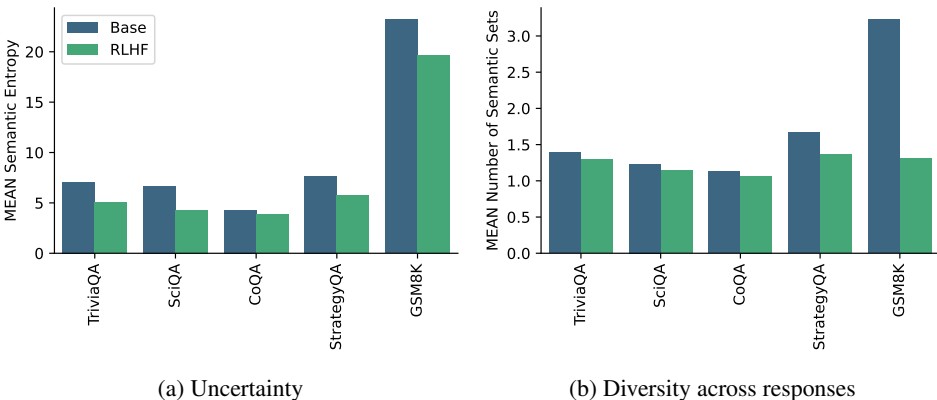

(a) Uncertainty                    (b) Diversity across responses

Figure 2: **RLHF finetuning leads to higher confidence and less diversity in responses.** These figures present a comparison between Llama2-70b pretrained and RLHF finetuned across various datasets. RLHF finetuned models exhibit (a) higher predictive confidence (lower mean predictive entropies across samples) and (b) are less diverse (lower number of semantic sets across samples) than pretrained models.

## 5.2 IN-DIALOGUE UNCERTAINTY CAN REDUCE HALLUCINATIONS FOR UNANSWERABLE QUESTIONS

Next we shift our focus to the common scenario in-dialogue-based chat where the model encounters questions it cannot answer, and therefore should abstain. We conduct an experiment to discern whether LLMs can differentiate between answerable questions, such as "What is a transformer architecture?" and logically unanswerable questions, such as "What color iPhone did Einstein prefer?".

In Table 2 we show predictive entropy and semantic entropy offer limited signal for distinguishing unanswerable questions in both pretrained and RLHF filetuned models. However, In-Dialogue Uncertainty as measured by AUROC is much more effective for distinguishing unaswerable questions. While In-Dialogue Uncertainty is more useful in both models with or without RLHF, we find RLHF finetuning enhances In-Dialogue Uncertainty awareness for unanswerable questions as shown in Table 2. We further probe this capability by illustrating RHLF results in far more hedge words in responses for unanswerable vs. unanswerable questions as shown in Appendix Figure 8 across

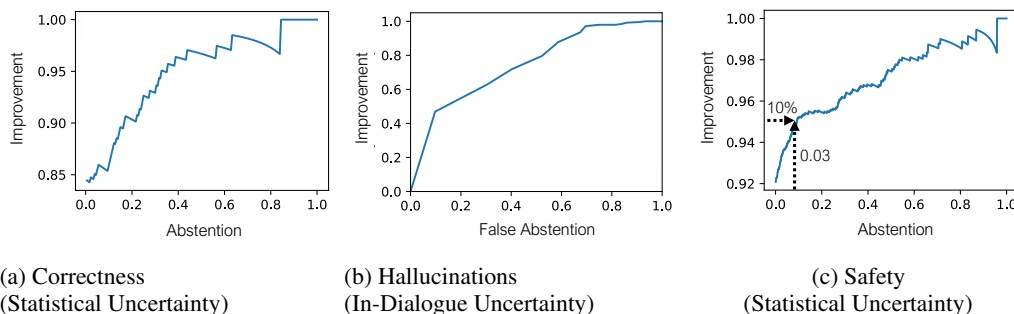

(a) Correctness
(Statistical Uncertainty)

(b) Hallucinations
(In-Dialogue Uncertainty)

(c) Safety
(Statistical Uncertainty)

Figure 3: **Uncertainty-based Abstention leads to improvements in correctness, hallucinations and safety when using the right uncertainty measures.** This figure shows Accuracy-Rejection Curves (ARCs) for correctness and safety and a Receiver Operating characteristics (ROC) curve with false abstention referring to the false positive rate for hallucination settings. Across all three scenarios, adopting an abstention approach for uncertain responses enhances accuracy w.r.t. correctness, improves the detection of unanswerable questions thereby reducing hallucinations, and boosts the safe response rate.

various model sizes. *These results suggests RLHF can reveal through In-Dialogue uncertainty when a question is unanswerable.*

**Reducing hallucinations for unanswerable questions with In-Dialogue uncertainty.** We now use In-Dialogue uncertainty to indicate when a model should abstain to reduce hallucinations for unanswerable questions. We show a simple filter for abstention based on In-Dialogue uncertainty can significantly reduce hallucinations for unanswerable questions. For example, In-Dialogue uncertainty can filter responses for nearly 50% of unanswerable questions at a cost of incorrectly refusing only 10% of answerable questions. We show the full receiver operating characteristic curve for all thresholds in Figure 3 b.

In real world applications where a user can ask different types of questions, we recommend that practitioners differentiate between different types of questions before they decide whether to use statistical uncertainty or verbalized uncertainty for abstention. We carry out a perplexity analysis of unanswerable questions and those seeking existing information and observe that a perplexity-based classifier differentiates sufficiently between those. Specifically, we compare 1000 randomly sampled questions from Science QA (answerable) versus SelfAware (unanswerable). We find as expected Science QA questions have a lower perplexity of 98.59 versus 108.07 for unanswered questions, suggesting perplexity could be a promising indicator to determine the answerability regime of a question.

| SelfAware Dataset | | AUROC w.r.t. | | | |
| --- | --- | --- | --- | --- | --- |
| | | Statistical Measures | | | InDU |
| | | Predictive Entropy | Semantic Entropy | Neg Log-Likelihood | |
| Answerable & unanswerable | Base | 0.64 | 0.60 | 0.62 | **0.69** |
| | RLHF | 0.59 | 0.60 | 0.48 | **0.75** |

Table 2: **In-Dialogue uncertainty (InDU) is better suited to distinguish between answerable questions and unanswerable questions in contrast to statistical uncertainty measures.** This table depicts AUROCs w.r.t. uncertainty metrics for answerable and unanswerable questions from the SelfAware dataset. RLHF finetuning leads to higher In-Dialogue Uncertainty AUROCs compared to the base model.

## 5.3 STATISTICAL UNCERTAINTY CAN IMPROVE THE SAFETY OF RLHF MODELS

Here we examine models' capacity to abstain from providing unsafe responses using uncertainty. We evaluate model responses to adversarial malicious prompts designed to elicit harmful responses using

two open-sourced datasets: AutoDAN (Zhu et al., 2023) and AttaQ (Kour et al., 2023). To ensure a fair comparison, we compare three models based on Llama2 with 7B parameters: pretrained only, instruction finetuned, and RLHF finetuned.

We find the pretrained model has the lowest safe response rates (7.4% for AttaQ and 9.0% for AutoDAN), followed by the supervised instruction finetuned model with response rates of 54.1% for AttaQ and 11.5% for AutoDAN. Finally, we find the Llama2 RLHF finetuned model exhibits the highest safe response rate of 92.1% for AttaQ and 92.5% for AutoDAN prompts. These results confirm instruction tuning and RLHF can improve the safety of model responses.

Next we analyze how well uncertainty measures can identify unsafe responses, measured via AUROC. As shown in Table 3, the pretrained Llama2 model exhibits random performance in identifying unsafe responses. Curiously, Semantic Entropy is particularly poor for AutoDAN, as the dataset is specifically constructed with the goal of preserving semantics of the original prompt (Zhu et al., 2023). Surprisingly, the supervised instruction finetuned model, although yielding a higher safe response rate, also mostly yields poor performance on both datasets. The RLHF finetuned Llama2 model on the other hand exhibits probability vectors that carry valuable information for properly identifying unsafe responses. As expected, In-Dialogue Uncertainty performs rather randomly, as while hedge words are used by the model to indicate unanswerable questions, they do not indicate unsafe responses. *Our experiments demonstrate that RLHF finetuning not only aligns the model to safety, but also notably enhances its uncertainty awareness w.r.t. safety.*

| Data | | AUROC w.r.t. | | | |
|------|------|---------------------|------------------|--------------------|------|
| | | Predictive Entropy | Semantic Entropy | Neg Log-Likelihood | InDU |
| AttaQ | Base | 0.57 | 0.53 | 0.43 | 0.48 |
| | Inst. Tune | 0.48 | 0.45 | 0.44 | 0.50 |
| | RLHF | **0.78** | **0.77** | **0.78** | **0.51** |
| AutoDAN | Base | 0.48 | 0.24 | 0.54 | 0.52 |
| | Inst. Tune | 0.50 | 0.33 | 0.68 | **0.67** |
| | RLHF | **0.96** | **0.94** | **0.99** | 0.41 |

Table 3: **Only RLHF finetuning leads to well performing AUROCs w.r.t. all statistical uncertainty metrics.** Evaluation of models using different alignment methods in distinguishing safe from unsafe responses on two adversarial datasets (AttaQ and AutoDAN). The RLHF finetuned Llama2 is evaluated against its pretrained version and a supervised instruction tuned variant, Vicuna. The AUROC is used to assess the model's ability in distinguishing safe from unsafe responses using a given uncertainty metric.

**Improving safety with statistical uncertainty in RLHF finetuned models.** We investigate how well uncertainty can determine when a model should abstain to improve safety. To do so, we compute the accuracy-rejection curve (ARC) where accuracy corresponds to whether a response is safe in Figure 3. For the RLHF model we observe that by abstaining to answer only the 10% most uncertain samples w.r.t. negative log-likelihood the proportion of safe responses for AttaQ improves from 92.1% to 95.1% and for AutoDAN even from 92.5% to 99.4%. We present additional results for ARC in the Appendix Figure 5. Furthermore, in Appendix 10 when investigating the ROC curve we find that across both adversarial datasets uncertainty can filter a majority of unsafe responses. For AutoDAN 99% of unsafe responses can be filtered out with only 10% falsely refused responses. For AttaQ, 70% of unsafe responses can be filtered while sacrificing 30% falsely refused responses. These results represent even further enhancements upon an already high-performing RLHF finetuned model in terms of safety. *Overall these findings demonstrate abstaining via statistical uncertainty can dramatically boost the safety of RLHF model responses.*

## 6 DISCUSSION

Uncertainty can be complicated in large language models. On the one hand, similar to classification, there is statistical uncertainty induced by the vector of token probabilities – together with its multiple versions, such as predictive and semantic entropy. On the other, there is In-Dialogue Uncertainty

induced by the presence of hedge words in responses. In addition, there are different forms of the models – with or without RLHF – that have different kinds of uncertainty profiles.

The main message of our study is that despite these complications, selecting the right kind of uncertainty together with the right form of the model can lead to better abstention decisions for question answering, in the process promoting correctness and safety, and reducing hallucinations for unanswerable questions. The form of uncertainty and the form of the model differ from one use-case to the next – statistical uncertainty works for correctness and safety, while in-dialogue uncertainty works for unanswerable questions. We have also introduced a new perplexity analysis to guide practitioners in selecting the appropriate uncertainty regime, enhancing the practical applicability of our findings for improving reliability. Specifically, we recommend leveraging statistical uncertainty in low-perplexity scenarios and in-dialogue uncertainty in high-perplexity settings.

Limitations of our work include that relying on the model's internal sense of uncertainty is inherently imperfect. Furthermore, we only discuss applications of hallucinations within the datasets available for unanswerable questions, but do not tackle all forms of possible hallucinations. Overall, our study underscores the importance of uncertainty in the context of LLMs, and we hope that it will contribute to the recognition of uncertainty as a critical factor in working with LLMs, as has been the case in other fields of machine learning for a long time (Hendrycks & Gimpel, 2018; Liang et al., 2020; Liu et al., 2021; Hendrycks et al., 2022).

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

APPENDIX

# A ACCURACY-REJECTION CURVES (ARCS) FOR CORRECTNESS AND SAFETY SCENARIOS

To provide deeper insights into the impact of uncertainty scores on the performance and safety of models, we present further Accuracy-Rejection Curves (ARCs) in the following sections. Generally, ARCs measure the accuracy on a subset of the dataset that is retained when samples are progressively rejected based on an uncertainty measure, from the most uncertain to the least uncertain. A higher Area Under the Accuracy-Rejection Curve (AUARC) indicates a more effective uncertainty measure, as it can better differentiate between incorrect and correct samples, or between safe and unsafe samples.

## A.1 CORRECTNESS SETTINGS

In the context of correctness settings, accuracy is defined as the ratio of correct samples to all samples in the remaining dataset. In Figure 4 we present ARC curves for both TriviaQA and SciQA, comparing the pretrained and RLHF finetuned models. For TriviaQA, the AUARC is identical for both models at 0.94. For SciQA, the RLHF finetuned model outperforms the pretrained model with a score of 0.83 versus 0.77. As we reject more samples, the accuracy increases, demonstrating that the use of uncertainty measures to reject samples effectively enhances the model's performance. This approach is particularly beneficial when the goal is to deploy a highly accurate model. While such a model may abstain more frequently, when it does provide an answer, it is more likely to be accurate.

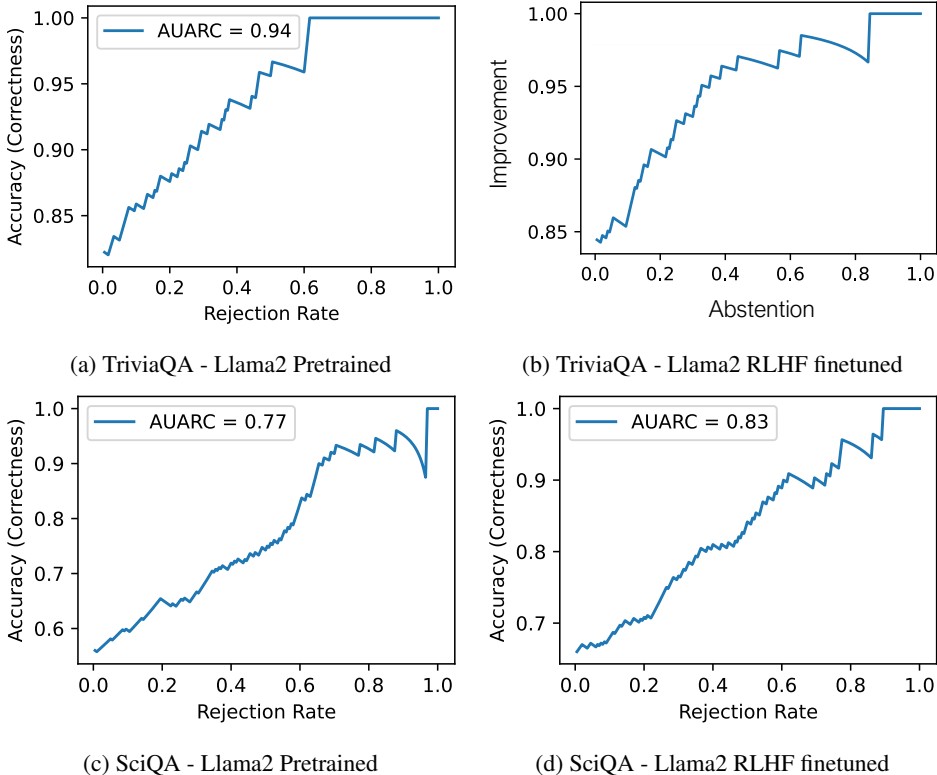

(a) TriviaQA - Llama2 Pretrained

(b) TriviaQA - Llama2 RLHF finetuned

(c) SciQA - Llama2 Pretrained

(d) SciQA - Llama2 RLHF finetuned

Figure 4: **Accuracy-Rejection Curves (ARCs) for TriviaQA and SciQA:** Accuracy is defined as the ratio of correct samples to all samples in the remaining dataset. Rejection rate denotes the proportion of progressively rejected samples based on the uncertainty measure.

## A.2    SAFETY SETTINGS

In safety settings, the accuracy in the ARC is defined as the ratio of safe samples to all samples in the remaining dataset. We use the keyword based safety metric to measure whether a response is safe or unsafe. We present ARC curves for AttaQ and AutoDAN, using the Llama2 70b RLHF finetuned model in Figure 5. The AUARC for AttaQ is 0.97, and for AutoDAN, it is near perfect at 0.99. Similar to the correctness setting, we observe that as more samples are rejected, the proportion of safe responses in the remaining dataset increases. This indicates that uncertainty metrics can be effectively employed to enhance the safety of models. In situations where particularly safe models are essential and a few unanswered questions can be tolerated, using uncertainty measures proves highly effective, as they require little to no additional computational effort.

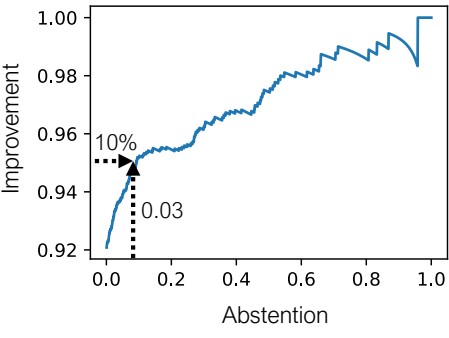
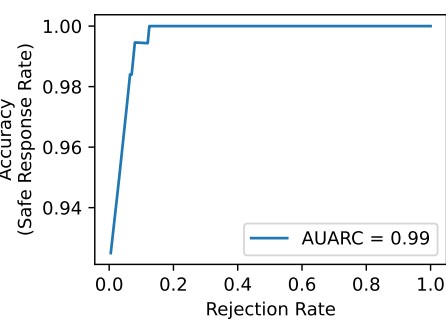

(a) AttaQ - Llama2 RLHF finetuned          (b) AutoDAN - Llama2 RLHF finetuned

Figure 5: **Accuracy-Rejection Curves (ARCs) for AttaQ and AutoDAN:** Accuracy is defined as the ratio of safe samples to all samples in the remaining dataset. Rejection rate denotes the proportion of progressively rejected samples based on the uncertainty measure.

## B    ADDITIONAL RESULTS FOR CORRECTNESS SETTINGS

This section presents additional results featuring more uncertainty metrics for fact-based Q&A settings. The datasets and prompting schemes used are identical to those in the main text. Figure 6 provides additional results for mean predictive entropy and mean negative log-likelihood, while Figure 7 presents the AUROC w.r.t. predictive entropy, semantic entropy and negative log-likelihood. These additional findings corroborate the results in the main paper, confirming that RLHF finetuning consistently leads to higher predictive confidence and less diversity in responses. However, the model's uncertainty awareness, as measured by AUROC, remains intact. We ran our experiments on a SLURM cluster with 8x32Gb Nvidia Tesla V100 GPUs for Llama2-70b (duration for all experiments 2 days).

## C    ADDITIONAL RESULTS FOR HALLUCINATION SETTINGS

Here we show further results for hallucination settings. Our findings in Figure 8 reveal that responses to unanswerable questions contain a higher average number of hedge words, indicating greater In-Dialogue Uncertainty, compared to responses to answerable questions. This pattern holds true across models of varying sizes and both with and without RLHF fine-tuning, although the difference in the number of hedge words in responses is slightly less pronounced for pretrained models. However, when it comes to mean statistical measures over samples, there is hardly any difference between answerable and unanswerable questions, which adversely impacts the AUROC as previously demonstrated. Moreover, the Receiver Operating Characteristic (ROC) curve with respect to In-Dialogue Uncertainty, as depicted in Figure 9, illustrates that InDU can effectively identify approximately 50% of unanswerable questions. This capability comes with a trade-off of incorrectly refusing only 10% of questions that could have been answered. We ran our experiments on a SLURM cluster with 8x32Gb Nvidia Tesla V100 GPUs for Llama2-70b and with 4x16Gb Nvidia Tesla V100 GPUs for Llama2-7b and Llama2-13b (duration for all experiments 1 day).

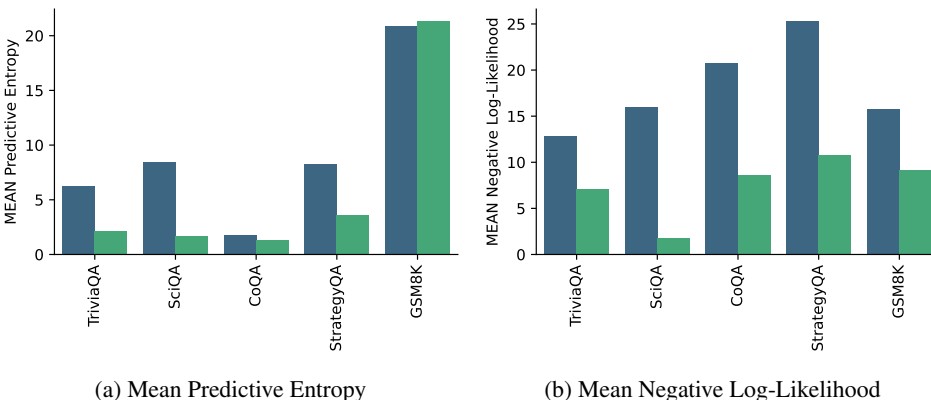

(a) Mean Predictive Entropy

(b) Mean Negative Log-Likelihood

Figure 6: **Comparison w.r.t. mean uncertainty metrics between Llama2-70b pretrained and RLHF finetuned across various datasets and prompting strategies (blue bars: Pretrained; green bars: RLHF finetuned):** Mean Predictive Entropy and mean Negative Log-Likelihood are both higher for pretrained models than for RLHF finetuned models.

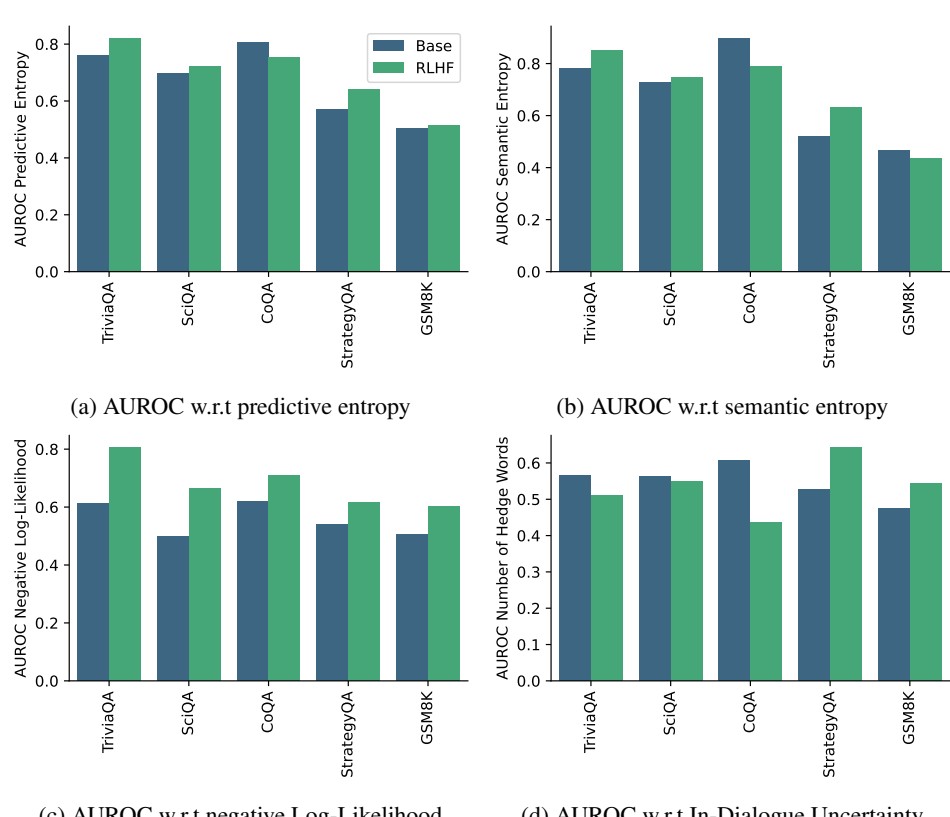

(a) AUROC w.r.t predictive entropy

(b) AUROC w.r.t semantic entropy

(c) AUROC w.r.t negative Log-Likelihood

(d) AUROC w.r.t In-Dialogue Uncertainty

Figure 7: **Comparison w.r.t. uncertainty awareness between Llama2-70b pretrained and RLHF finetuned across various datasets (blue bars: Pretrained; green bars: RLHF finetuned):** Both models are alternately better than the other regarding the level of uncertainty awareness, as indicated by the AUROC w.r.t. different uncertainty metrics. The RLHF fine-tuned model only demonstrates a slight advantage for AUROC w.r.t. negative log-likelihood.

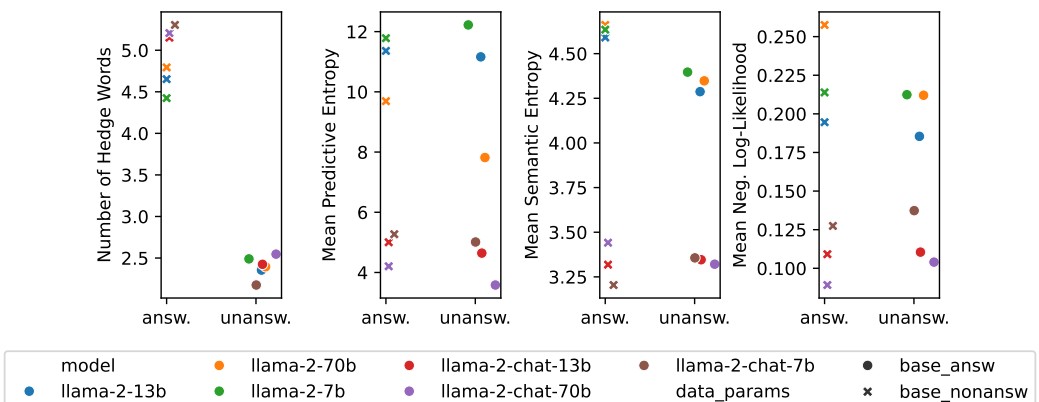

Figure 8: **In-Dialogue Uncertainty (InDU) shows a much higher gap between answerable and unanswerable questions than statistical measures.** Statistical uncertainty measures exhibit almost the same average scores for answerable and unanswerable questions. In contrast, In-Dialogue Uncertainty, based on the number of hedge words, shows a clear difference, with responses to unanswerable questions containing twice as many hedge words as responses to answerable questions. These results remain largely consistent regardless of model size and whether or not RLHF finetuning is applied.

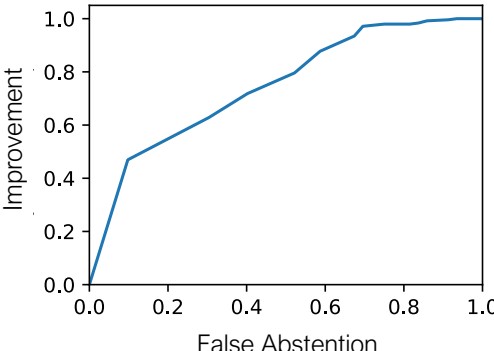

Figure 9: **In-Dialogue Uncertainty enables the filtering of 50% of unanswerable questions at the cost of incorrectly refusing 10% of answerable questions.** This figure illustrates the Receiver Operating Characteristic (ROC) curve for distinguishing between answerable and unanswerable questions using In-Dialogue Uncertainty.

## D    ADDITIONAL RESULTS FOR SAFETY SETTINGS

We further investigate the Receiver Operating Characteristic (ROC) curves of all three models, which are based on Llama2 with 7B parameters: pretrained only, instruction finetuned, and RLHF finetuned. The models are evaluated on two datasets: AutoDAN (Zhu et al., 2023) and AttaQ (Kour et al., 2023). In Figure 10 we find that for the RLHF finetuned model on AutoDAN almost 100% of unsafe responses can be filtered out with almost no falsely refused responses. For AttaQ, 70% of unsafe responses can be filtered while sacrificing 30% falsely refused responses. We ran our experiments on a SLURM cluster with with 4x16Gb Nvidia Tesla V100 GPUs (duration for all experiments  12h).

## E    COMBINING HALLUCINATION & CORRECTNESS

Here we combine the hallucination with the correctness setting to inverstigate whether the model can even differentiate between answerable questions ("What is a transformer architecture?") that are answered incorrectly due to a lack of specific knowledge, and logically unanswerable questions

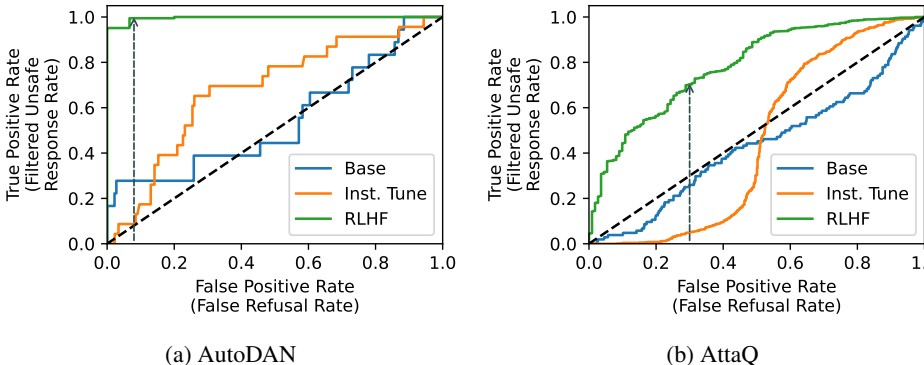

(a) AutoDAN

(b) AttaQ

Figure 10: **Unsafe responses can be filtered out reliably based on uncertainty scores.** The figures depict ARC curves for two different redteaming datasets evaluating the ability of the RLHF finetuned Llama2-7b to detect unsafe responses using negative log-likelihood. On the AutoDAN dataset, the model can filter out 100% of unsafe responses with almost no falsely refused responses. For the AttaQ dataset, the model can filter out 70% of unsafe responses at the cost of a 30% false refusal rate.

("What color iPhone did Einstein prefer?"). As shown in Table 4, we measure the AUROC between falsely answered and unanswerable questions and find that statistical uncertainty measures perform no better than random. Surprisingly, In-Dialogue Uncertainty performs nearly as well on this task as in distinguishing answerable from unanswerable questions. A more detailed analysis in Table 5 reveals that statistical uncertainty measures are, on average, very similar for incorrectly answered questions and unanswerable questions. Only In-Dialogue Uncertainty is twice as high for unanswerable questions than for incorrectly answered questions. This demonstrates that LLMs are capable of distinguishing not only between answerable and unanswerable questions but also between wrongly answered questions and unanswerable questions. However, instead of expressing this distinction through log probabilities, LLMs convey it through the use of hedge words.

| Data Configuration for SelfAware | Mode | AUROC w.r.t. | | | |
| | | Statistical Measures | | | InDU |
| | | Predictive Entropy | Semantic Entropy | Neg Log-Likelihood | |
| Answerable (False) & unanswerable | Base | 0.50 | 0.47 | 0.59 | **0.66** |
| Answerable (False) & unanswerable | RLHF | 0.46 | 0.47 | 0.44 | **0.72** |
| Answerable & unanswerable | Base | 0.64 | 0.60 | 0.62 | **0.69** |
| Answerable & unanswerable | RLHF | 0.59 | 0.60 | 0.48 | **0.75** |

Table 4: **In-dialogue verbalized uncertainty can distinguish between incorrectly answered questions and unanswerable questions, a scenario where statistical measures show no indicative power.** This table depicts AUROCs w.r.t. uncertainty metrics for incorrectly answered questions and unanswerable questions from the SelfAware dataset. RLHF finetuning leads to higher in-dialogue verabalized uncertainty AUROCs compared to the base model.

## F    FURTHER INSIGHTS INTO IN-DIALOGUE UNCERTAINTY

To gain deeper insights into the In-Dialogue Uncertainty, we examine the hedge words used in the model's responses. Figure 11 compares incorrect responses from the pretrained with those from the RLHF finetuned model. We find that the RLHF finetuned model exhibits greater diversity in expressing verbalized uncertainty within dialogues. Furthermore, Figure 12 reveals a significant difference in the number of hedge words used in responses to answerable and unanswerable questions. Specifically, a large proportion of responses to answerable questions contain no hedge words, whereas responses to unanswerable questions typically include at least one hedge word. The mean number of hedge words in responses to answerable questions is 1.67 (with a standard deviation of 2.3). In contrast, the mean number of hedge words in responses to unanswerable questions is nearly three

| Data Configuration for SelfAware | Mode | Mean | | | |
|---|---|---|---|---|---|
| | | Statistical Measures | | | InDU |
| | | Predictive Entropy | Semantic Entropy | Neg Log-Likelihood | |
| Answerable (False) | RLHF | 4.63 | 6.96 | 11.2 | 1.92 |
| Unanswerable | | 4.20 | 6.62 | 10.6 | 4.79 |
| Answerable (False) | Base | 10.34 | 10.60 | 25.3 | 1.75 |
| Unanswerable | | 9.69 | 9.90 | 28.0 | 4.41 |

Table 5: **Our In-Dialogue Uncertainty (InDU) measure shows a notable difference between incorrectly answered and unanswerable questions.** This table presents the mean uncertainty measures for both answerable and unanswerable questions. While the statistical uncertainty measures exhibit only a slight increase for unanswerable questions compared to incorrectly answered questions, the mean number of hedge words for unanswerable questions is more than twice that for incorrectly answered questions.

times higher, at 4.79 (with a standard deviation of 4.45). These findings suggest that the natural use of hedge words in model responses can serve as an indicator of whether a question is answerable or unanswerable, as evidenced by the high AUROC values reported in the main paper.

# G    VALIDITY OF FUZZY EXACT MATCH

To evaluate the correctness of model responses, we derive a ground truth label via utilizing fuzzy exact match. This method, unlike the traditional exact match metric that necessitates a word-for-word match between the response and the reference answer, determines if the reference answer is contained within the response. This approach is more resilient to variations in model response styles, while preserving the interpretability inherent in exact match.

We validate the effectiveness of fuzzy exact match by comparing it with human evaluations on 200 samples each from TriviaQA and SciQA. We juxtapose the accuracy derived from these human evaluations with that obtained from fuzzy exact match, as well as the widely used exact match and F1 score metrics based on RougeL. For TriviaQA, the accuracy based on human evaluations is 87.5%, closely mirrored by the 85.0% accuracy of fuzzy exact match. In contrast, the accuracies based on exact match and F1 score are significantly lower, at 14.5% and 27.8%, respectively. A similar trend is observed for SciQA, with human evaluations yielding an accuracy of 77.0%, fuzzy exact match 65.0%, and exact match and F1 score only 2.5% and 7.0% respectively. These findings indicate that the commonly used metrics of exact match and F1 score diverge significantly from human evaluations when assessing the correctness of responses. On the other hand, fuzzy exact match aligns much more closely with human evaluations and is still equally interpretable. Consequently, we adopt fuzzy exact match as our metric for correctness throughout our study.

# H    VALIDITY OF SAFETY METRICS

We employ two metrics to compute the safe response rate of models given a dataset: one is based on Llama Guard (Inan et al., 2023), and the other utilizes the keyword-based approach described in Zhu et al. (2023); Zou et al. (2023). The safe response rate is defined as the ratio of safe responses to all responses. To validate these metrics, we compare them with human evaluations on the AutoDAN dataset (Zhu et al., 2023). Each response is assessed by three human judges, and a majority vote determines the safety of the response. The safe response rate for human evaluations is 92.5%, identical to the keyword-based metric, as the two sets are the same. However, Llama Guard yields a safe response rate of 100% on this dataset. We further conduct a pairwise t-test for Llama Guard, resulting in a p-value of $8.35e-10$. Consequently, we use the keyword-based metric in the main paper and present results for Llama Guard on AttaQ in the appendix in Table 6.

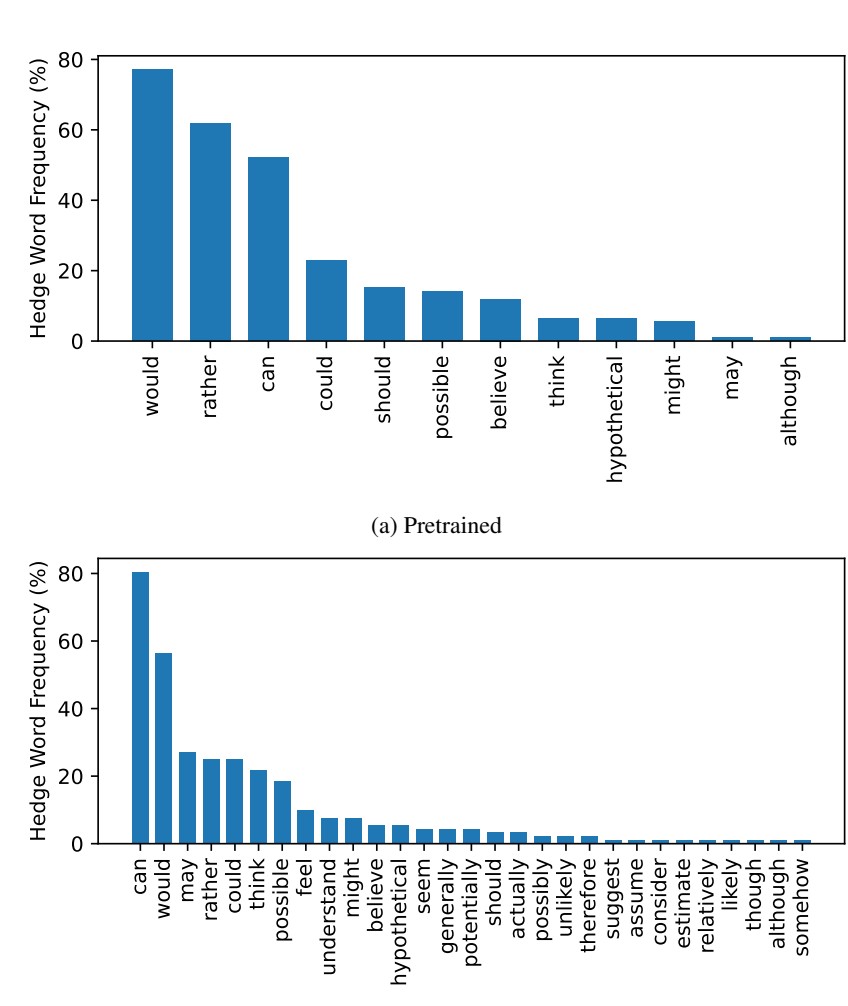

(a) Pretrained

(b) RLHF finetuned

Figure 11: **Hedge words in responses between a Llama2 pretrained and RLHF finetuned model:** Llama2-70b pretrained uses different hedge words compared to the RLHF finetuned version in responses to known unknown (unanswerable) questions. The RLHF finetuned Llama2-70b is more diverse in expressing verbalized uncertainty in-dialogue.

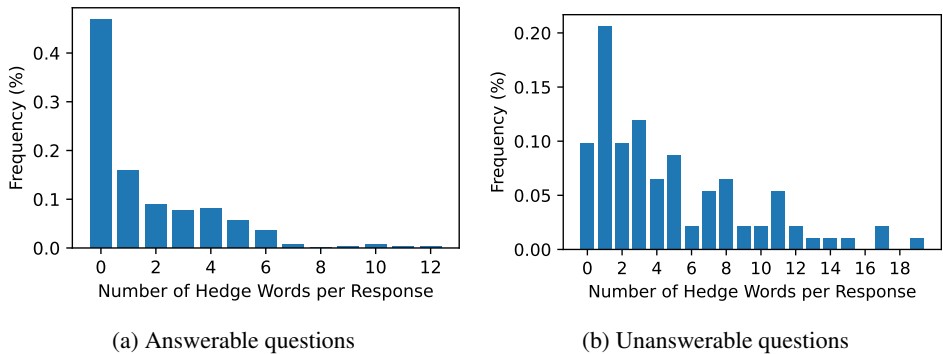

(a) Answerable questions

(b) Unanswerable questions

Figure 12: **Number of hedge words in responses between answerable and unanswerable questions:** Llama2-70b RLHF finetuned exhibits more hedge words for unanswerable questions than it does for answerable questions.

| Data | Mode | AUROC w.r.t. | | | |
|------|------|-----------------------|-----------------|----------------------|------|
| | | Predictive Entropy | Semantic Entropy | Neg Log-Likelihood | VIDU |
| AttaQ | Base | 0.56 | 0.53 | 0.50 | 0.46 |
| | Inst. Tune | 0.44 | 0.45 | 0.29 | 0.35 |
| | RLHF | **0.79** | **0.52** | **0.98** | **0.92** |

Table 6: **Only RLHF finetuning leads to well performing AUROCs w.r.t. uncertainty metrics, underscoring that pretraining and supervised instruction tuning are insufficient for inducing safety-related uncertainty awareness into the model. Safety of responses is calculated with Llama Guard.** Evaluation of models using different alignment methods in distinguishing safe from unsafe responses on AttaQ. The RLHF finetuned Llama2-7b is evaluated against its pretrained version and a supervised instruction tuned variant, Vicuna. The AUROC is used to assess the model's ability in distinguishing safe from unsafe responses.

