# OpenReview forum: "Uncertainty-Based Abstention in LLMs Improves Safety and Reduces Hallucinations"
_ICLR.cc/2025/Conference — Submitted to ICLR 2025_

### Official Review · Reviewer_5sjZ · 2024-11-02

**Soundness:** 3
**Presentation:** 3
**Contribution:** 3
**Rating:** 5
**Confidence:** 4

**Summary:**

This paper proposes uncertainty-based methods for abstaining from providing incorrect, hallucinated, or unsafe answers. It considers probabilistic uncertainty (log likelihood, entropy, and semantic uncertainty) as well as verbal uncertainty. Experiments with Llama2 models across various question-answering and adversarial-prompting benchmarks demonstrate that (1) the considered uncertainty measures contain information about whether an answer is incorrect, hallucinated, or unsafe, and (2) abstention based on these measures is effective.

**Strengths:**

- Studied topic is timely and important.
- Paper outlines multiple applications of the proposed method.
- Paper is well-written and easy to follow.
- Experiments test diverse uncertainty measures
- Experiments consider different model sizes and variants.

**Weaknesses:**

- There exist more sophisticated approaches to these problems but they are not compared empirically [1, 2]. [1] build classifiers based on hidden representations, and [2] constructs ensembles using LLM prompting. Authors mention the weaknesses of prompting and fine-tuning in related work but do not demonstrate them through experiments. In fact, the experiments do not seem to concern distribution shift so there is no reason not to compare with those methods.
- Related work is missing some recent work showing similar results (e.g., [1,2,3]).
- Experiment sections mostly discuss observations but does not attempt to explain the observed phenemena.
- Some parts of the experiment section are unclear or can be further improved. Specifically, in figure 3, "statistical uncertainty" should be replaced with a specific measure and model (e.g., entropy). It is also missing model names. The plots need to have baseline curves to clearly illustrate improvements.

[1] https://arxiv.org/abs/2304.13734

[2] https://arxiv.org/abs/2402.00367

[3] https://arxiv.org/abs/2402.13213

**Questions:**

"we recommend that practitioners differentiate between different types of questions before they decide whether to use statistical uncertainty or verbalized uncertainty for abstention..." Could you explain why the experiment you conducted supports this claim?

---

### Official Review · Reviewer_NpjB · 2024-11-02

**Soundness:** 2
**Presentation:** 2
**Contribution:** 2
**Rating:** 5
**Confidence:** 4

**Summary:**

The authors investigate the potential of uncertainty-based abstention to improve performance in question-answering tasks, specifically focusing on correctness, hallucinations, and safety scenarios. They analyze two types of uncertainty—statistical uncertainty and in-dialogue uncertainty—and examine the effects of RLHF on these uncertainties.

**Strengths:**

1. The motivation behind this work is compelling and relevant for the deployment and real-world application of LLMs. However, while the authors highlight the benefits of combining RLHF with uncertainty to enhance performance, reviewers suggest that additional validation experiments, particularly in the areas of hallucination and safety, would strengthen the claims.
2. The paper underscores the importance of uncertainty for abstention, demonstrating that incorporating uncertainty can improve various aspects of model performance.

**Weaknesses:**

1. Although the reviewers appreciate the study's motivation, they raise concerns regarding the experimental setup. For instance, in the hallucination settings, tests are only conducted on the SelfAware dataset. It would be beneficial to include additional datasets to more comprehensively evaluate the method's effectiveness in reducing hallucinations, especially given that current approaches primarily rely on Retrieval-Augmented Generation (RAG) [1].

2. In the safety setting, the reviewers are interested in seeing how the uncertainty mechanism performs across a broader range of evaluation datasets. For example, PKU-SafeRLHF [3] provides safe, decoupled preferences and red-teaming prompts; how does the proposed approach perform on safety measures in these rigorous evaluations via case by case gpt-4 evaluation?

3. The reviewers are not fully convinced by the claim that "our experiments demonstrate that RLHF fine-tuning not only aligns the model with safety but also enhances its uncertainty awareness in relation to safety." RLHF alone does not guarantee model safety, particularly when the preference data distribution is uncertain. For instance, the GPT-4 technical report highlights that while RLHF helps align model responses with user intent, models may still exhibit brittle or undesired behaviors on both safe and unsafe inputs, especially when labeler instructions during reward model data collection are underspecified. Reviewers suggest that the authors provide a more detailed discussion on this aspect and include comparisons with models specifically designed for safety alignment, such as RLCD [2] and Safe RLHF [3].

4. Regarding evaluation, the authors rely primarily on statistical measures, such as keyword-based approaches. However, this static evaluation method may fall short of detecting nuanced harmful responses, such as those involving emotional abuse. Additionally, Llama Guard’s performance drops in non-OOD (Out-of-Distribution) scenarios. Reviewers recommend including case-by-case GPT-4 evaluations to directly assess the safety of two responses, providing a more granular safety evaluation.

[1] Self-RAG: Learning to Retrieve, Generate, and Critique through Self-Reflection
[2] RLCD: Reinforcement Learning from Contrastive Distillation for Language Model Alignment
[3] PKU-SafeRLHF: Towards Multi-Level Safety Alignment for LLMs with Human Preference

**Questions:**

See above.

**Details Of Ethics Concerns:**

This paper utilizes human evaluations to conduct a safety assessment of the model's outputs. The authors state, "We validate the effectiveness of fuzzy exact match by comparing it with human evaluations on 200 samples each from TriviaQA and SciQA." However, details regarding the background and diversity of these 200 individuals remain unclear, as well as whether these evaluations comply with IRB requirements.

---

### Official Review · Reviewer_KHqs · 2024-11-02

**Soundness:** 2
**Presentation:** 3
**Contribution:** 2
**Rating:** 3
**Confidence:** 5

**Summary:**

This paper focuses on abstention and uncertainty in LLMs, benchmarking how useful different uncertainty estimates are across three broad tasks.
These tasks are correctness, unanswerable questions, and safety.
Correctness is evaluated against standard QA data (TriviaQA, SciQA, CoQA, StrategyQA, and GSM8K).
Unanswerable vs answerable questions are sourced from the SelfAware dataset and SciQA.
Adversarial examples are sourced from AttaQ and AutoDAN.
The authors examine negative log-likelihood, predictive entropy, semantic entropy, and In-Dialogue Uncertainty, which is the number of hedge tokens present in the output.
All experiments were run on Llama2.
Across different tasks, the authors find that different uncertainty estimates lead to better or worse calibration, with no one method consistently outperforming the others.
The authors show that thresholding uncertainty scores can lead to better correctness, safety, and less hallucination on unanswerable questions.

**Strengths:**

- **Well written and clearly organized**: the paper is easy to follow, the writing is clear, and the questions being tested are clear.
- **In-dialogue uncertainty metric is new**: As far as I can tell, past work has not proposed counting the number of hedge words as a method of confidence estimation.
- **Sufficient datasets examined**: The authors do a good job of testing multiple datasets to make their point.

**Weaknesses:**

- **Limited novelty:** The novelty of the paper is pretty limited. From the abstract/intro, it seems like the main contribution of the paper is in showing that abstention based on uncertainty can improve results. This result is not new (see next point about missing related work). Moreover, the primary methodological novelty in this work is In-Dialogue Uncertainty, which is a fairly small contribution and does not consistently provide benefits in all settings. The Discussion presents a more nuanced view of the contribution (i.e. framing this paper as a survey of confidence estimation methods and showing that there isn't one method that consistently does well.) This framing would have been more novel but then I would have expected to see more different uncertainty estimation methods tested.
- **Missing related work:** This paper misses a large chunk of the related work on abstention and confidence estimation from the last 2 years, focusing on older work. Examples:
	- https://arxiv.org/pdf/2407.18418
	- https://arxiv.org/abs/2308.13387
	- https://arxiv.org/abs/2311.09677
	- https://arxiv.org/abs/2404.00474
	- https://arxiv.org/abs/2405.21028
	- https://arxiv.org/abs/2401.06730
	- https://aclanthology.org/2024.naacl-long.301/

- **Outdated models**: It's not clear why the authors only conduct experiments on Llama2, when there are many newer and more performant models available (even in the same family). To make a strong claim about when different estimation methods work and don't work, I would have expected to see more open-source models tested.
- **No unified method**: one way this paper could have been made more compelling is if it presented a unified estimation method/recipe that worked well across settings. Currently, the paper does not have any such unified method.

**Questions:**

- It would be worth discussing the tradeoff between abstention and usability further.
- In-Dialogue Uncertainty is given an acronym but the acronym isn't used. It's also misspelled on L053.

---

### Official Review · Reviewer_XzXp · 2024-11-04

**Soundness:** 3
**Presentation:** 3
**Contribution:** 1
**Rating:** 3
**Confidence:** 5

**Summary:**

Paper shows that abstention based on different measures of uncertainty for different types of prompts works well. Specifically, for correctness and safety, statistical uncertainty-based abstention helps improve correctness and reduce unsafe responses. For hallucinations, abstention based on in-dialogue uncertainty (coined by authors as the inclusion of phrases such as "I don't know" in model responses) helps reduce hallucinations.

**Strengths:**

- Well-written paper with a clear walkthrough over the different problems and uncertainty metric considerations
- Interesting idea of using in-dialogue uncertainty as a measure of response uncertainty
- Clear description of experiments, metrics, and results; strong scientific method

**Weaknesses:**

- It should not come as a surprise that using uncertainty metrics helps LLMs abstain when they should not engage with the prompt, as shown in Kadavath (2022) and multiple other papers cited in the related works. The core contributions of this paper can be boiled down to the introduction InDU (which also was inspired by an existing paper by Islam (2020)) and when to use each kind of uncertainty, both of which seem more fitting for, e.g., an appendix in Kadavath's paper, especially since this reads more like a survey paper of implementation details than novel ideas or concepts
- Minor: various typos such as "In-Dialogoe" in Introduction, Islam et al. without year in 3.2

**Questions:**

- How can we practically account for all possible hedge words for every use case? Some prompts might even require responses to include hedge words; seems like a lot of finetuning and engineering effort to incorporate this uncertainty metric
- I'm not sure I agree with hallucinations being only considered for unanswerable questions. LLMs definitely hallucinate in other situations. How extendable are these findings?
- Statistical uncertainty metrics perform at more or less the same level. What should the reader take away from all these results?

---

### Official Review · Reviewer_SxZN · 2024-11-05

**Soundness:** 4
**Presentation:** 4
**Contribution:** 2
**Rating:** 3
**Confidence:** 5

**Summary:**

The paper studies how token-level and semantic uncertainty metrics on generated LLM text relate to accuracy on knowledge-intensive tasks, hallucination on unanswerable questions, and response safety on adversarial/malicious question datasets. Among the uncertainty metrics explored is one based on on counting hedge words in model responses. Experiments show that these uncertainty metrics are useful for model abstention in order to improve correctness of model generations, reduce hallucination, and increase response safety (at the cost of an increased abstention rate). Experiments are conducted across many relevant datasets using Llama 2 models.

**Strengths:**

- Important: The core idea of the paper is sensible, relating uncertainty metrics to abstention in order to improve factuality and safety of model responses.
- Important: The experiment design is sound and the chosen metrics are reasonable. The experiments include many relevant datasets for measuring knowledge, hallucination, and safety.
- Of some importance: The paper is fairly well-written and easy to follow.

**Weaknesses:**

- Very important: The novelty of the work is quite limited in my view. The high level conclusion that uncertainty is useful for abstention has already been thoroughly explored. What I see as this paper’s contributions beyond this observation are: (1) measuring in-dialogue uncertainty can help with abstention, specifically for factuality/hallucination; (2) uncertainty can help with safety, as it turns out that responses on AutoDAN-like datasets are more likely to be unsafe if they are uncertain. I don’t think the paper claims much beyond this. So a further issue with the novelty here is that (1) has already been shown, more or less, in https://arxiv.org/abs/2405.21028. The (2) result is interesting but I do not think it is a large enough result for a full paper, and it is not explored in much depth beyond one paragraph in this paper.
- Important: The measurement of in-dialogue uncertainty, even if useful, is a heuristic that does not feel particularly generalizable, especially compared to other model-based measurements of in-dialogue confidence.

**Questions:**

- To be clear, in these experiments, the model might not actually abstain, right? You would have to calculate these metrics and then hide the response from the user if it were deemed unacceptable, right?
- It was hard to tell if there was a proposed training or inference method from reading the intro. It took me a while to realize that this was more of an analysis paper, showing how these metrics could be used for filtering model outputs.
- Sec. 3 probably doesn’t need to take up as much space is it currently does (people should know what NLL is), but at the same time it could give more understanding into the metrics (computing actual entropy over samples is hard, so you compute predictive entropy).
- L.377 typo “unanswerable vs. unanswerable”
- L.471 “abstaining to answer”→ “abstaining from answering”

---

### Meta-Review · Program_Chairs · 2024-12-24

**Metareview:**

PC is entering meta-review on behalf of SAC/AC:

The reviewers did not believe that this paper was a strong contribution given the limited novelty of the work, and lack of anchoring in the field.

**Additional Comments On Reviewer Discussion:**

TBD

---

### Decision · Program_Chairs · 2025-01-22

Reject